# Predictors of puerperal menstruation

George Uchenna Eleje[1,2]*, Emmanuel Onyebuchi Ugwu[3], Victor Okey Dinwoke[4], Perpetua Kelechi Enyinna[4], Joseph Tochukwu Enebe[4], Innocent Igwebueze Okafor[4], Livinus Nnanyere Onah[4], Osita Samuel Umeononihu[1], Chukwudi Celestine Obiora[4], Sylvester Onuegbunam Nweze[4], Ekene Agatha Emeka[5], Chinekwu Sochukwu Anyaoku[5], Frank O. Ezugwu[4]

1 Department of Obstetrics and Gynecology, Nnamdi Azikiwe University Teaching Hospital, Nnewi, Nigeria, 2 Effective Care Research Unit, Department of Obstetrics and Gynecology, Faculty of Medicine, Nnamdi Azikiwe University, Awka, Nigeria, 3 Department of Obstetrics and Gynecology, College of Medicine, University of Nigeria, Nsukka, Nigeria, 4 Department of Obstetrics and Gynecology, ESUT Teaching Hospital, Parklane, Enugu, Nigeria, 5 Department of Family Medicine, Nnamdi Azikiwe University Teaching Hospital, Nnewi, Nigeria

* georgel21@yahoo.com, gu.eleje@unizik.edu.ng

**Data Availability Statement:** All relevant data are within the manuscript and its Supporting Information files.

**Funding:** The author(s) received no specific funding for this work.

## Abstract

### Background

Puerperal period is an important and thought-provoking period for puerperal mothers. Surprisingly, reports have indicated that there is increasing number of women resuming menstruation within six weeks of childbirth (puerperal menstruation). To the best of knowledge, there is no prior study on predictors of puerperal menstruation.

### Objective

To determine frequency and predictors of puerperal menstruation.

### Methods

This was a single tertiary health institution cross-sectional study at ESUT Teaching Hospital, Parklane, Enugu, Nigeria that included data from May 2015 to December 2018. Women were interviewed at the end of the first six weeks of their childbirth. Women with HIV positive or had uterine rupture or peripartum hysterectomy were excluded. Bivariate analysis was performed by the chi-squared test and conditional logistic regression analysis was used to determine variables associated with puerperal menstruation. Statistical significance was accepted when *P*- value is <0.05.

### Results

A total of 371 women met the inclusion criteria. The return of menses within 6 weeks was present in 118(31.8%) women versus 253 (68.2%) women without puerperal menstruation, given a ratio of 1:3. Of the 371 women, 249 (67.1%) were on exclusive breastfeeding. The significant associated risk factors were age (p = 0.009), parity (p<0.001), early use of family planning (p = 0.001), socio-economic status (p<0.001) and manual removal of placenta (p = 0.007). At conditional logistic regression analysis, early use of family planning (p = 0.001),

**Competing interests:** The authors have declared
that no competing interests exist.

exclusive breastfeeding (p = 0.027) and manual removal of placenta (p = 0.012) were inde-
pendently associated with puerperal menstruation. Induction/augmentation of labor, postpar-
tum misoprostol use and mode of delivery were not statistically significant (p>0.05, for all)

## Conclusion

One in 3 women resumes menstruation within 6 weeks of childbirth. The major predictor
was early initiation of family planning, and exclusive breastfeeding with manual removal of
placenta a major protective factor. These interesting issues require further investigation to
better understand the mechanism of puerperal menstruation.

## Introduction

Puerperal period is an important and thought-provoking period in the life of puerperal moth-
ers. It is a period that receives relatively less attention than pregnancy and delivery and such
neglect may trigger unintended pregnancy [1, 2]. This is because, women are frequently
fecund postpartum before they realize it [2]. Thus, if a woman in her puerperal period has
resumed coital activity and is not on any effective method of family planning, she may be at
risk of becoming pregnant before her resumption of menstruation [2, 3].

Four mechanisms related to the resumption of normal menstrual cycles after childbirth have
been described in the literature [4, 5]. While the first two (weaning and infant mortality) have a
direct causal effect, the third (breastfeeding patterns of women who menstruated while they were
still breastfeeding) and fourth (maternal nutrition and health status) may have indirect causal
effects [4, 5]. The physiological mechanism underlying these observed relations originates in the
hypothalamus [6] and the pituitary gland [7], which all could be influenced by external factors
[8–11]. Thus, the longer the return of menses is delayed, the more likely it is that ovulation will
precede menses return and lack of menstruation does not preclude initiation of ovulation [12].

Published studies indicate that among women practicing postpartum abstinence, irregular
sexual activity may occur early, progressing to regular coital activity later [13, 14]. Individual
studies have drawn linkages between return of menses and initiation of contraceptives [13, 14].
Adanikin et al found that family planning use is most likely in the month following menses
return especially male condom use and withdrawal method [14]. By extrapolation, the data in
Nigeria is alarming because, the overall levels of contraceptive use are low in Nigeria since in
the 2008 national demographic health survey (NDHS), 15 percent of currently married women
were using any contraceptive method, and only 10 percent were using a modern method [15].
Currently, Nigeria has set a goal of a 36% contraceptive prevalence rate by 2018 [6].

In low-income country settings, a significant number of women experience pregnancy
complications such as hypertensive disorders of pregnancies and miscarriages. These lead to
fetal wastage or death of the newborn. Data from a study in Iran showed that, on average, child
survival increases the duration of postpartum amenorrhea and subsequent fall in postpartum
sexual abstinence [16]. In Nepal, the median duration of post-partum amenorrhea was more
than twice lower among women whose children died early compared to those without child
loss [17]. Additionally, a few HIV positive women avoid breastfeeding and opt for exclusive
breast milk substitutes to prevent mother to child transmission. All these factors may affect the
duration of postpartum amenorrhea. Although, there have been previous studies on postpar-
tum practices among parturients, none has studied the timing of initiation of menses in a
Nigerian population [1, 18]. Again, anecdotal reports have indicated that there is increasing

number of women resuming menstruation within six weeks of childbirth irrespective of exclusive breastfeeding [personal communication]. Furthermore, a recent Nigerian study has demonstrated that the uptake of modern contraception is very poor especially in the postpartum period [19]. The consequence of all these is an increasing prevalence of short inter-birth interval in the study population with its attendant risks on maternal and neonatal health [19]. Therefore, studying the duration of puerperal amenorrhea and the factors associated with the return of menses will help in patients counseling as well as in designing policies/strategies aimed at increasing the duration of inter-birth interval. This study therefore is aimed at determining the frequency and predictors of resumption of menstruation within the first six weeks of childbirth (puerperal menstruation).

## Materials and methods

The study was conducted in postnatal care clinic at the ESUT Teaching Hospital, Parklane, Enugu, Nigeria. This is a state tertiary hospital that manages all issues relating to pregnant women from within and outside the state. The study included postnatal women seen in the clinic between 1st May 2015 and 31st December 2018. At the time of postnatal clinic visits, the women received brief information about the study and were invited to participate in the study by the researchers. Written informed consent was obtained from all individual participants.

This is a cross-sectional descriptive study using an interviewer-administered structured pretested questionnaire on the women's socio-demographic characteristics (age, marital status, parity, booking status, and educational status, retroviral status, etc), postpartum activities (time at resumption of menses and coitus postpartum), use of contraceptives (whether natural or modern method), induction of breastfeeding, perineal tear/ episiotomy at child birth, and type of peri-partum activities of the women. Any woman who has attended and registered at least one antenatal clinic prior to delivery or labor is said to be booked.

The natural family planning or fertility awareness included the method of contraception that does not use any drugs or devices. It included any of the calendar or rhythm method, cervical mucus method or the basal body temperature method [3].The following steps were involved in the questionnaire development and validation. Firstly, the questionnaire's face validity was established by having it reviewed by two different experts (obstetrician-gynecologist and measurement and evaluation expert in test construction). The obstetrician-gynecologist ensured that the questions successfully captured the topic on puerperal menstruation. The measurement and evaluation expert ensured that our questionnaire did not contain common errors such as confusing, leading, or double-barreled questions. Secondly, we ran a pilot test by selecting 40 of our intended postpartum women. These helped us to weed out weak or irrelevant questions. Thirdly, we cleaned the collected data, and fourthly, we used principal analysis which validated what the questionnaire was actually measuring and fifthly we revised the questionnaire based on the information we gathered from the analysis.

Social class stratification was determined in accordance to Olusanya et al [20]: classes 1, 2, and 3 were considered high class, and classes 4 and 5 considered low social class. Tertiary education was defined as polytechnic or university education. The sample size was obtained using the formula [21] $N = Z^2 alphaPQ/d^2$ where: Z = standard normal deviation at 95% confidence interval; P = prevalence of the problem (median prevalence rate for return of menses by 6 months postpartum in a recent Cochrane review by Van der Wijden and Manion was put at 25.3% [5]); Q = 1-p and d = 0.05. The ultimate was adjusted to allow a non-inferiority sample size of 289 obtained and rounded up to 347 to cater for 20% attrition or non-response.

All eligible and consenting women seen at the end of the first six weeks of their childbirth were recruited. The eligible women were consecutively recruited from post-natal clinic of the

hospital. Pregnant women or HIV positive women, including those that had uterine rupture or hysterectomy or secondary postpartum hemorrhage, or who were less than six weeks ($\leq$5th week) postpartum or women more than six weeks ($\geq$7th week) postpartum at the time of interview were excluded.

Ethical clearance for this study was obtained from the research and publication ethics committee of ESUT Teaching Hospital, Parklane, Enugu, Nigeria. Three doctors including the researchers were involved in the interviewing and collection of data from the women on arrival at study site. In order to maintain the absolute confidentiality of the respondents, there were no identifying marks on the questionnaires.

Data was entered after checking completeness, cleaning and coding into computer Epi info version 7.0 (Centers for Disease Control and Prevention, USA) and the results were displayed in tables. To determine the relationship between puerperal menstruation and childbirth, chi-squared test, Fisher's exact test and t-test whenever appropriate, were performed in the bivariate analysis. Conditional logistic regression was employed in the multiple regression analysis to determine variables associated with puerperal menstruation, while controlling for other confounding variables (age, booking status, parity, mode of delivery, socio-economic class and gestational age at delivery). In this analysis, the odds ratio and confidence interval was set at 95% and p<0.05 was considered significant.

## Results

Three hundred and seventy one respondents met the inclusion criteria and their questionnaires were correctly filled. The mean age of the respondents was 29.5±5.1 years (range = 20–44 years). All the respondents were married and majority, 174 (46.9%) have tertiary level of education. The socio-demographic characteristic of the respondents' is shown in Table 1.

All the respondents were currently breastfeeding their babies. Of the 371 women, 118 (31.8%) had their menses resumed within the first 6 weeks of childbirth while 253 (68.2%) women did not resume menstruation within 6 weeks of childbirth.

The association between the resumption of menses and respondents' socio-demographic and peripartum characteristics is shown in Table 2. The gestational age at delivery, oxytocin augmentation in labor/induction of labor, route of delivery, evacuation of retained product of conception and misoprostol insertion after child birth had no significant association with resumption of menses at 6 weeks postpartum (p > 0.05).

Of the 118 women with puerperal menstruation, 71 (60.2%) were already on family planning method, while 41 (16.2%) of the 253 women without puerperal menstruation were on family planning. Of the women on family planning methods, 18 (25.4%) of 71 women with puerperal menstruation were on modern family planning methods, while 9 (22.0%) of the 41 women without puerperal menstruation were on modern family planning methods.

Regarding puerperal menstruation, the menstrual group and the non-menstrual group were compared using multiple logistic regression, while controlling the effects of possible confounding variables, such as the age, booking status, parity, mode of delivery, socio-economic class and gestational age at delivery. This is shown in Table 3. The results showed that the risk of puerperal menstruation approximately was two times higher in those women who had early initiation of family planning compared to those who did not [(P<0.001), odds ratio (OR) (95% confidence interval (CI) = 2.07 (1.59 to 2.70)].

## Discussion

The present study is the first study in Nigeria, which examined the frequency and predictors of puerperal menstruation. Remarkably, our results revealed that the frequency of puerperal

**Table 1. Socio-demographic characteristics of the respondents.**

| Variables | Frequency (N = 371) | Percentage |
|---|---|---|
| **Age** | | |
| 20–24 | 70 | 18.9 |
| 25–29 | 112 | 30.2 |
| 30–34 | 137 | 36.9 |
| 35–39 | 41 | 11.1 |
| 40–44 | 11 | 2.9 |
| **Booking Status** | | |
| Booked | 296 | 79.8 |
| Un-booked | 75 | 20.2 |
| **Parity** | | |
| 1 | 123 | 33.2 |
| 2–4 | 184 | 49.5 |
| ≥5 | 64 | 17.3 |
| **Educational level** | | |
| Primary | 39 | 10.5 |
| Secondary | 158 | 42.6 |
| Tertiary | 174 | 46.9 |
| **Social Class** | | |
| I | 13 | 3.5 |
| II | 33 | 8.9 |
| III | 30 | 8.1 |
| IV | 193 | 52.0 |
| V | 102 | 27.5 |

menstruation was 31.81% and younger age, multiparity, early use of family planning; high socio-economic status, exclusive breastfeeding and manual removal of placenta had a significant relationship with the early return of menses within 6 weeks of childbirth.

These findings are noteworthy. For instance individual studies have drawn linkages between menses return and initiation of contraceptive use [3, 14]. Adanikin et al in their study found that family planning use is most likely in the month following menses return [14]. However, the finding is in contrast to a recent Cochrane review where median prevalence of 25.3% of breastfeeding women were seen to have resumed menstruation by six months postpartum, but not six weeks postpartum [5]. This is despite the fact that majority were said to be practicing exclusive breastfeeding.

The findings also highlighted that exclusive breastfeeding was associated with puerperal menstruation, even though majority of the women in the puerperal menstruation group were practicing exclusive breastfeeding. This is despite the fact that prolonged lactation expectedly suppresses the production of certain types of hormones, thereby extending the postpartum anovulatory period. This finding is not consistent with findings of studies in different countries [4, 22–24]. Although this study did not assess the frequency and intensity of sucking of the infant, it may have been suboptimal in the majority of women studied.

In the previous studies, the death of the index child during infancy was associated with the early return of menstruation [16, 17]. This too is not consistent with findings of our study. In our study, all the respondents had living index child. Ideally, it is the death of a child during infancy that cuts short the duration of breast feeding, which results in earlier resumption of menses and ovulation.

**Table 2. Association between resumption of menses and respondents' characteristics based on bivariate test.**

| Variables/subgroup | Menses Group (N = 118) | No menses group (N = 253) | P-value |
|---|---|---|---|
| Mean Age | 25.9±3.9 years | 29.6±5.5 years | *0.009 |
| Booking Status | | | |
| Booked | 89 (75.4) | 207 (81.8) | 0.154 |
| Unbooked | 29 (24.6) | 46 (18.2) | |
| Parity | | | |
| Primiparous | 18 (15.3) | 105 (41.5) | *<0.001 |
| Multiparous | 100 (84.7) | 148 (58.5) | |
| Socio-economic Class | | | |
| High | 59 (50.0) | 17 (6.7) | *<0.001 |
| Low | 59 (50.0) | 236 (93.3) | |
| Mode of Delivery | | | |
| Vaginal | 83 (70.3) | 195 (77.1) | 0.164 |
| Cesarean section | 35 | (29.7) | 58 (22.90 |
| Episiotomy/ Perineal tear | | | |
| Yes | 65 (55.1) | 136 | (53.8) |
| 0.811 | | | |
| No | 53 (44.9) | 117 (46.2) | |
| Oxytocin augmentation/ induction | | | |
| Yes | 47 (39.8) | 99 (39.1) | 0.898 |
| No | 71 (60.2) | 154 (60.9) | |
| Misoprostol use after child birth | | | |
| Yes | 35 (29.7) | 70 (27.7) | 0.692 |
| No | 83 (70.3) | 183 (72.3) | |
| Manual removal of placenta | | | |
| Yes | 5 (4.2) | 34 (13.4) | *0.007 |
| No | 113 (95.8) | 219 (86.6) | |
| Evacuation of retained product of conception | | | |
| Yes | 11 (9.3) | 11 (4.3) | 0.059 |
| No | 107 (90.7) | 242 (95.7) | |
| Gestational age at Delivery | | | |
| ≥37 weeks | 95 (80.5) | 189 (74.7) | 0.220 |
| <37 weeks | 23 (19.5) | 64 (25.3) | |
| Coitus | | | |
| Yes | 59 (50.0) | 83 (32.8) | 0.279 |
| No | 59 (50.0) | 170 (67.2) | |
| Exclusive Breastfeeding | | | |
| Yes | 89 (75.4) | 160 (63.2) | * 0.020 |
| No | 29 (24.6) | 93 (36.8) | |

The effects of mode of delivery, booking status and gestational age at delivery were not significant. The risk of resumption of menses for women who had cesarean section was lower than those that had vaginal delivery, though this too did not reach statistically significant difference. Nevertheless, younger age, multiparity, early use of family planning and high socio-economic status had a significant relationship with the early return of menses within the first 6 weeks of childbirth. The shorter duration for return of menses among high socio-economic women was perhaps due to the fact that the educational level and employment status were higher in them

**Table 3. Association between resumption of menses and respondents' postpartum activities based on bivariate test and multiple logistic regression.**

| Variables/subgroup | Menses Group | No menses group | Unadjusted | OR (95% CI) | Adjusted | OR (95%CI) |
|---|---|---|---|---|---|---|
| | (N = 118) | (N = 253) | P-value | | P-value | |
| **Manual removal of placenta** | | | | | | |
| Yes | 5 (4.2) | 34 (13.4) | *0.007 | 0.29 (0.14–0.55) | *0.012 | 0.38 (0.13–0.57) |
| No | 113 (95.8) | 219 (86.6) | | Reference | | Reference |
| **Family planning** | | | | | | |
| Yes | 71 (60.2) | 41 (16.2) | *0.001 | 2.73 (1.89–3.94) | *0.001 | 2.07 (1.59–2.70) |
| No | 47 (39.8) | 212 (83.8) | | Reference | | Reference |
| **Exclusive Breastfeeding** | | | | | | |
| Yes | 89 (75.4) | 160 (63.2) | *0.020 | 1.06 (0.77–1.46) | *0.027 | 1.08 (0.78–1.49) |
| No | 29 (24.6) | 93 (36.8) | | Reference | | Reference |

Conditional logistic regression was employed (P<0.1) in the multiple regression analysis to control confounding variables: age, booking status, parity, mode of delivery, socio-economic class and gestational age at delivery.

Abbreviations: 95%CI = 95% Confidence interval; OR = Odds ratio.

than women of low socio-economic group. This finding agrees with a previous published report by Aryal who revealed that educated women had a 1.5 times higher risk of returning menstruation early compared to their uneducated counterpart [17]. The exact explanation for this finding is difficult. However, it is possible that the respondents of higher socio-economic class have high nutritional status which has been shown to influence early return of menses [4, 5]. Furthermore, it is possible that most educated and employed mothers did not have enough time to breastfeed their children (despite being on exclusive breastfeeding mode) as they work outside and, thus, tend to lactate for a shorter period and also probably provide food supplements to their children much earlier. Further studies are necessary to confirm this assumption.

With regard to age and parity, the early return of menses was seen more in younger aged women and in women with high parity. Therefore, women who may be most affected are those of our younger women folk who have given birth to two or more children at their young age. Interestingly, Aryal in 2010 had come to similar conclusions in his study of Nepalese women [17]. Aryal report revealed that younger mothers are most likely to terminate breastfeeding early as compared to older counterparts and this invariably will potentiate their early return of their menstruation. The risk of resumption of menses for urban women was higher than rural women in one study [4]. This may also explain the early return of menses in some groups because the location of the study hospital is urban [4].

Current use of contraception in Nigeria has increased from 6 percent of currently married women in 1990 to 13 percent in 2003 and 15 percent in 2008. Currently, Nigeria government has set a goal of a 36% contraceptive prevalence rate by 2018 [6]. There has been a corresponding increase in the use of modern contraceptive methods, from 4 percent in 1990 to 8 percent in 2003 to 10 percent in 2008 [15]. Thus, in this study, the use of the modern method of family planning was seen in 25.8% of the women with puerperal menstruation versus 22.0% without puerperal menstruation. This finding agrees with previous study by Ezebialu and Eke in Nnewi, Nigeria where 21.5% of mothers used a modern family planning method during the early postpartum period [25]. Modern methods of contraception may be viewed with suspicion especially within six weeks of childbirth. This may explain why more than 70% of women on family planning methods as seen in this study practiced natural form of family planning.

The findings of this interpretative study failed to confirm that initiation of coital activity within 6 weeks of childbirth is highly influenced by the return of menses. This is because, although

50.0% of women that resumed menstruation within 6 weeks of childbirth have already commenced coital activity in contrast to only 32.8% in women whose menses have not resumed, such difference failed to reach statistical significance (p>0.05). In all, the rate of resumption of coital activity within six weeks of childbirth in the women studied was higher than the previous report by Ezebialu and Eke in Nnewi, Nigeria where they reported a rate of only 29.7% [25].

Interestingly, this study reveals that initiation family planning method within 6 weeks of childbirth could be highly influenced by the early return of menses. This is because 60.2% of women that had puerperal menstruation were already on family planning method. In contrast, only 16.2% of women that have not resumed menstruation within six weeks were on family planning and this difference was statistically significant (p = 0.001). This finding is understandable and may not be novel as research has consistently shown that women who were menstruating regularly are more likely to use contraception in the postpartum period because resumption of menses is a signal to return of ovulation and high possibility of pregnancy [3, 14]. The early return of menses is again associated with other possible explanatory variables in many ways in the study subjects due to the diverse socio-economic and intrapartum events such as manual removal of placenta.

It is intriguing to observe the association between the manual removal of placenta and decreased incidence of puerperal menstruation. As long as we know, our study appears to be the first to report such potential association showing that women that underwent manual removal of placenta have significantly decreased incidence of puerperal menstruation. This finding is not surprising as manual removal of the placenta can also be a risk factor for acute postpartum endometritis [26]. However, when there is established puerperal menstruation in women with recent history of successfully managed retained placenta, the history of delayed hemorrhage or postpartum endometritis must be ruled out [26].

The main limitation of our study is lack of information on nutritional status as this is likely to influence the amenorrhea in the postpartum period. Although we excluded all women who were ≥7 weeks postpartum, the potential for recall bias among the respondents cannot be completely ruled out. The risk factors may be associated with each other and so, further analysis with adjustment may give some more insight. Another limitation, is the small probability that some of these women may have continued with postpartum (delayed) bloody lochia, and did not presented a real menstrual period which follows the ovulation two weeks before, especially when most women were breastfeeding. However, this study tried to exclude such bloody lochia. Also, there was a long interval of data collection which may have introduced bias of double counting in the study. This study is a single center study and so a multi-center, multi-regional study is needed for future studies on the topic. Such future study should also assess the frequency and intensity of sucking of the infant in the breastfeeding women.

## Conclusion

In conclusion, our data show that approximately one third of the parturients achieve puerperal menstruation and the significant associated risk factors include younger age, multiparity, early use of family planning, high socio-economic status, 'exclusive breastfeeding' and manual removal of placenta. Of these, the major predictor was early initiation of family planning, while exclusive breastfeeding with manual removal of placenta was a major protective factor. These study findings could be useful to understand among the clinicians and patients the importance to increase prevention strategies aimed at avoiding unintended pregnancies during the postpartum period, especially in younger women. This study has raised interesting issues and requires further investigation to better understand the mechanism of puerperal menstruation.

## Supporting information

**S1 Data.**
(XLSX)

## Acknowledgments

The current work had taken great efforts from all colleagues that work in the ESUT Teaching Hospital, Enugu, Nigeria, who kindly participated in the questionnaire distribution. Great thanks for all who shared and helped to put this work in its final form.

## Author Contributions

**Conceptualization:** George Uchenna Eleje, Emmanuel Onyebuchi Ugwu, Victor Okey Din-woke, Perpetua Kelechi Enyinna, Frank O. Ezugwu.

**Data curation:** Joseph Tochukwu Enebe, Innocent Igwebueze Okafor, Livinus Nnanyere Onah, Frank O. Ezugwu.

**Formal analysis:** Innocent Igwebueze Okafor, Osita Samuel Umeononihu, Chukwudi Celes-tine Obiora, Ekene Agatha Emeka, Chinekwu Sochukwu Anyaoku.

**Investigation:** Livinus Nnanyere Onah.

**Methodology:** George Uchenna Eleje, Emmanuel Onyebuchi Ugwu, Victor Okey Dinwoke, Perpetua Kelechi Enyinna.

**Project administration:** George Uchenna Eleje, Victor Okey Dinwoke, Innocent Igwebueze Okafor, Sylvester Onuegbunam Nweze, Frank O. Ezugwu.

**Supervision:** George Uchenna Eleje, Emmanuel Onyebuchi Ugwu, Victor Okey Dinwoke, Perpetua Kelechi Enyinna, Joseph Tochukwu Enebe, Innocent Igwebueze Okafor, Livinus Nnanyere Onah, Sylvester Onuegbunam Nweze, Frank O. Ezugwu.

**Validation:** Osita Samuel Umeononihu, Chukwudi Celestine Obiora, Ekene Agatha Emeka, Chinekwu Sochukwu Anyaoku.

**Visualization:** George Uchenna Eleje, Emmanuel Onyebuchi Ugwu, Victor Okey Dinwoke, Perpetua Kelechi Enyinna, Sylvester Onuegbunam Nweze, Frank O. Ezugwu.

**Writing – original draft:** Victor Okey Dinwoke, Joseph Tochukwu Enebe, Osita Samuel Umeononihu, Chukwudi Celestine Obiora.

**Writing – review & editing:** George Uchenna Eleje, Emmanuel Onyebuchi Ugwu, Victor Okey Dinwoke, Perpetua Kelechi Enyinna, Joseph Tochukwu Enebe, Sylvester Onuegbu-nam Nweze, Ekene Agatha Emeka, Chinekwu Sochukwu Anyaoku, Frank O. Ezugwu.

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
