## [Decision Letter · Decision Letter 0]

23 Mar 2020

PONE-D-19-23514

Predictors of puerperal menstruation

PLOS ONE

Dear Dr. ELEJE,

Thank you for submitting your manuscript to PLOS ONE. After careful consideration, we feel that it has merit but does not fully meet PLOS ONE’s publication criteria as it currently stands. Therefore, we invite you to submit a revised version of the manuscript that addresses the points raised during the review process.

The manuscript has been assessed by two reviewers; their comments are available below.

The reviewers have raised a number of major concerns that need attention in a revision, the reviewers note that further information should be reported about the recruitment of participants and the statistical analyses undertaken, and they raise that the lack of information on nutritional status is a major limitation as this is likely to influence the amenorrhea postpartum period.

In addition to the items raised by the reviewers, I have the following concerns which I ask you also address during your revision:

Please provide further information under the Methods section on how the questionnaire was developed and validated prior to its use in this study. Please also include a copy of the questionnaire with the revised manuscript, in its original language and also a translation if this was not in English.The study is based on a single center, the manuscript needs to provide some further justification on how the population involved is representative of the population of women in the country.This statement in the Conclusions is not supported and must be deleted of revised ‘*The acceptance of family planning was highly influenced by the early resumption of menses although the acceptance or use of modern family planning methods is low’* the study is cross-sectional by design and as a result a temporal link cannot be established between family planning and early resumption of menses, the study can only establish associations.The findings regarding exclusive breastfeeding require clarification, the abstract refers to exclusive breastfeeding as a major predictor, however, the discussion indicates ‘breastfeeding was not associated with a lower risk for return of menstruation’.

Could you please carefully revise the manuscript to address the comments raised?

Please note that the revised manuscript will need to undergo further review, we thus cannot at this point anticipate the outcome of the evaluation process.

We would appreciate receiving your revised manuscript by May 07 2020 11:59PM. Please include the following items when submitting your revised manuscript:

We look forward to receiving your revised manuscript.

Kind regards,

Iratxe Puebla

Deputy Editor-in-Chief, PLOS ONE

Journal Requirements:

Please ensure that your manuscript meets PLOS ONE's style requirements, including those for file naming. The PLOS ONE style templates can be found at http://www.plosone.org/attachments/PLOSOne_formatting_sample_main_body.pdf and http://www.plosone.org/attachments/PLOSOne_formatting_sample_title_authors_affiliations.pdf

Reviewers' comments:

Reviewer's Responses to Questions

**Comments to the Author**

1. Is the manuscript technically sound, and do the data support the conclusions?

Reviewer #1: Partly

Reviewer #2: Yes

2. Has the statistical analysis been performed appropriately and rigorously? 

Reviewer #1: No

Reviewer #2: Yes

3. Have the authors made all data underlying the findings in their manuscript fully available?

Reviewer #1: No

Reviewer #2: Yes

4. Is the manuscript presented in an intelligible fashion and written in standard English?

Reviewer #1: Yes

Reviewer #2: Yes

5. Review Comments to the Author

Reviewer #1: Method

The authors have not paid attention to the work and there are a number of inconsistencies in the work. For example, in the sample size calculation, the authors indicated on page 7 line 19 that they used 32.3% of women who return to menses after 6 months, however on page 8 line 3, they indicated that they used 25.3%.

Again the method section looks scanty and lacks details of the sampling method used. Though, it is stated that "all eligible and consenting women seen at the end of the first six weeks of their childbirth were recruited" (line 6 page 8), It is hard to believe that in a tertiary hospital that manages all issues relating to pregnant women from within and outside the state, only 371 postnatal mothers were recruited over a 3 and half year period. How was the recruitment done?

Again, it is stated that participants "were recruited until the calculated sample size was reached". How then was it possible for the sample size of 347 to be overachieved (371)?

The authors need to check whether they used multiple logistic regression or multivariate logistic regression. The results in Table 3 suggest a binary multiple logistic regression was used as opposed to multivariate logistic regression.

Results:

There are several inconsistencies in the results presented few are shown below:

Parity: Table 1 shows 253 women were multiparous, however, Table 2 shows 248 (100 + 148).

Booking: 283 booked (table 1) compared 296 (table 2)

Social Class: 200 high (table 1) compared to 76 (Table 2)

Discussion

On page 14 line 16 and 17, the authors indicated that "what matters most is the frequency and intensity of sucking of the infant which may be suboptimal in the majority of women studied", however, no data was presented in the results section on "frequency and intensity of sucking of the infants". What then is the basis for that assertion?

Reviewer #2: In general I think the paper has a strong justification, taking into account the low prevalence of family planning in this country and probably the high prevalence of unmet need for postpartum family planning. The conclusions could be useful to understand among the clinicians and patients the importance to increase prevention strategies to avoid unintended pregnancies during the postpartum period, especially in younger patients.

Regarding the content, I have some minor comments:

• Page 4, line 2: Grammar mistake: To add “The” Puerperal mothers and remove “the” before nursing mothers

• Page 4, line 20: The acronym FP is not explain in the previous or in the subsequent paragraphs, it is well known that it refers to Family Planning but it is better to clarify.

• Page 5, line 15-16: The statement “there have been previous studies on postpartum practices among parturients, none has studied the timing of initiation of menses“applies only for Nigeria or worldwide? If only applies for Nigeria, would be better to clarify.

• Page 8, line 21: What do you refer with booking status? Could you please clarify

• The lack of variable nutritional status is a limitation of this study because it has a direct influence on the amenorrhea postpartum period.

• Another limitation, is the small probability that some of these women have continued with postpartum loquios, and did not presented a real menstrual period, maybe would be good to clarify which is this probably according other studies.

• Within the discussion I recommend to extend the analyses about the association between the manual removal of placenta and return of menses.

Thank you very much,

6. PLOS authors have the option to publish the peer review history of their article (what does this mean?). If published, this will include your full peer review and any attached files.

Reviewer #1: No

Reviewer #2: No

---

## [Author Response · Author response to Decision Letter 0]

2 Apr 2020

01-04-2020

From 

Corresponding author (George Eleje)

To 

Editor

PLOS ONE

Dear editors:

Re: Submission of Response to Reviewers’ comments on Manuscript ID PONE-D-19-23514 entitled “Predictors of puerperal menstruation”.

Please find enclosed a point-by-point response (and highlights in yellow and in track changes in the manuscript) to the comments by the editors. We hope that the editors and reviewers will find the revisions acceptable.

REVIEWER COMMENT

• Dear Dr. ELEJE,

Thank you for submitting your manuscript to PLOS ONE. After careful consideration, we feel that it has merit but does not fully meet PLOS ONE’s publication criteria as it currently stands. Therefore, we invite you to submit a revised version of the manuscript that addresses the points raised during the review process.

The manuscript has been assessed by two reviewers; their comments are available below.

The reviewers have raised a number of major concerns that need attention in a revision, the reviewers note that further information should be reported about the recruitment of participants and the statistical analyses undertaken, and they raise that the lack of information on nutritional status is a major limitation as this is likely to influence the amenorrhea postpartum period.

AUTHORs’ RESPONSE

Many thanks for the commendations and thank you for finding merit to our manuscript. We have addressed ALL the concerns adequately. We have appended under limitations as follows: Our lack of information on nutritional status is a major limitation as this is likely to influence the amenorrhea postpartum period.

REVIEWER COMMENT

In addition to the items raised by the reviewers, I have the following concerns which I ask you also address during your revision:

• Please provide further information under the Methods section on how the questionnaire was developed and validated prior to its use in this study. Please also include a copy of the questionnaire with the revised manuscript, in its original language and also a translation if this was not in English.

AUTHORs’ RESPONSE

• Many thanks for the remarks. The following steps were involved in the questionnaire development and validation: Firstly, the questionnaire’s face validity was established by having it reviewed by two different experts (obstetrician-gynecologist and measurement and evaluation expert in test construction). The obstetrician-gynecologist ensured that the questions successfully captured the topic on puerperal menstruation. The measurement and evaluation expert ensured that our questionnaire did not contain common errors such as confusing, leading, or double-barreled questions. Secondly, we ran a pilot test by selecting 40 of our intended postpartum women. These helped us to weed out weak or irrelevant questions. Thirdly, we cleaned the collected data, and fourthly, we used principal components analysis which validated what the questionnaire was actually measuring and fifthly we revised the questionnaire based on the information you gathered from the principal components analysis.

• WE have appended under method section of the manuscript as follows:

• The following steps were involved in the questionnaire development and validation: Firstly, the questionnaire’s face validity was established by having it reviewed by two different experts (obstetrician-gynecologist and measurement and evaluation expert in test construction). The obstetrician-gynecologist ensured that the questions successfully captured the topic on puerperal menstruation. The measurement and evaluation expert ensured that our questionnaire did not contain common errors such as confusing, leading, or double-barreled questions. Secondly, we ran a pilot test by selecting 40 of our intended postpartum women. These helped us to weed out weak or irrelevant questions. Thirdly, we cleaned the collected data, and fourthly, we used principal analysis which validated what the questionnaire was actually measuring and fifthly we revised the questionnaire based on the information we gathered from the analysis.

• 

• REVIEWER COMMENT

• The study is based on a single center, the manuscript needs to provide some further justification on how the population involved is representative of the population of women in the country. 

AUTHORs’ RESPONSE

Many thanks for the remarks. Although, it is a single center, the city is centrally located with available international airport and receives referral from neighboring states and beyond. The hospital is a major referral tertiary health institution in eastern Nigeria. It is dominated by three major tribes in Nigeria. However, a multi-center, multi region study is needed for future studies on the topic. We have appended it as one of the limitations of the study as follows: This study is a single center study and so a multi-center, multi-regional study is needed for future studies on the topic.

• REVIEWER COMMENT

• This statement in the Conclusions is not supported and must be deleted of revised ‘The acceptance of family planning was highly influenced by the early resumption of menses although the acceptance or use of modern family planning methods is low’ the study is cross-sectional by design and as a result a temporal link cannot be established between family planning and early resumption of menses, the study can only establish associations.

AUTHORs’ RESPONSE

• Many thanks for the remarks. As suggested, the entire statement has been deleted in the conclusion part of the manuscript.

• 

• REVIEWER COMMENT

• The findings regarding exclusive breastfeeding require clarification, the abstract refers to exclusive breastfeeding as a major predictor, however, the discussion indicates ‘breastfeeding was not associated with a lower risk for return of menstruation’.

AUTHORs’ RESPONSE

• Many thanks for the remarks. We have now clarified the discussion statement on exclusive breastfeeding and that of the abstract. This is now appended as follows: “The findings also highlighted that exclusive breastfeeding was associated with puerperal menstruation, even though majority of the women in the puerperal menstruation group were practicing exclusive breast feeding. This is despite the fact that prolonged lactation expectedly suppresses the production of certain types of hormones, thereby extending the postpartum anovulatory period. This finding is not consistent with findings of studies in different countries [4, 22-24]

 REVIEWER COMMENT

Could you please carefully revise the manuscript to address the comments raised?

 AUTHORs’ RESPONSE

We have revised the manuscript and have addressed the comments adequately. The details are still running as seen below.

REVIEWER COMMENT

Please note that the revised manuscript will need to undergo further review, we thus cannot at this point anticipate the outcome of the evaluation process.

AUTHORs’ RESPONSE

• Many thanks for the remarks. We strongly anticipate favorable response or approval.

REVIEWER COMMENT

We would appreciate receiving your revised manuscript by May 07 2020 11:59PM. AUTHORs’ RESPONSE

Many thanks for the remarks.

REVIEWER COMMENT

AUTHORs’ RESPONSE

We do not wish to make further changes in our financial disclosure.

REVIEWER COMMENT

• A rebuttal letter that responds to each point raised by the academic editor and reviewer(s). This letter should be uploaded as separate file and labeled 'Response to Reviewers'.

• A marked-up copy of your manuscript that highlights changes made to the original version. This file should be uploaded as separate file and labeled 'Revised Manuscript with Track Changes'.

• An unmarked version of your revised paper without tracked changes. This file should be uploaded as separate file and labeled 'Manuscript'.

AUTHORs’ RESPONSE

• Many thanks for the remarks. We have included the three requested documents.

 REVIEWER COMMENT

Please note while forming your response, if your article is accepted, you may have the opportunity to make the peer review history publicly available. The record will include editor decision letters (with reviews) and your responses to reviewer comments. If eligible, we will contact you to opt in or out. We look forward to receiving your revised manuscript.

 AUTHORs’ RESPONSE

Many thanks for the remarks. 

Kind regards,

Iratxe Puebla

Deputy Editor-in-Chief, PLOS ONE

AUTHORs’ RESPONSE

Thank you.

Journal Requirements:

Please ensure that your manuscript meets PLOS ONE's style requirements, including those for file naming. The PLOS ONE style templates can be found at http://www.plosone.org/attachments/PLOSOne_formatting_sample_main_body.pdf and http://www.plosone.org/attachments/PLOSOne_formatting_sample_title_authors_affiliations.pdf

AUTHORs’ RESPONSE

Reviewers' comments:

Reviewer's Responses to Questions

Comments to the Author

1. Is the manuscript technically sound, and do the data support the conclusions?

Reviewer #1: Partly

Reviewer #2: Yes

 AUTHORs’ RESPONSE

Many thanks. It has also improved with the revision.

2. Has the statistical analysis been performed appropriately and rigorously?

Reviewer #1: No

Reviewer #2: Yes 

AUTHORs’ RESPONSE

We have improved on the statistical analysis.

 REVIEWER COMMENT

3. Have the authors made all data underlying the findings in their manuscript fully available?

Reviewer #1: No

Reviewer #2: Yes

 AUTHORs’ RESPONSE

 We have now included the questionnaires with the resubmission.

4. Is the manuscript presented in an intelligible fashion and written in standard English?

Reviewer #1: Yes

Reviewer #2: Yes

 AUTHORs’ RESPONSE

Many thanks for your commendation.

REVIEWER COMMENT

5. Review Comments to the Author

AUTHORs’ RESPONSE

Many thanks.

REVIEWER COMMENT

Reviewer #1: Method

The authors have not paid attention to the work and there are a number of inconsistencies in the work. For example, in the sample size calculation, the authors indicated on page 7 line 19 that they used 32.3% of women who return to menses after 6 months, however on page 8 line 3, they indicated that they used 25.3%.

AUTHORs’ RESPONSE

Many thanks for the remarks. We apologize for the inconsistencies. We have deleted the issue in the sample size calculation, that the authors indicated on page 7 line 19 that they used 32.3% of women who return to menses after 6 months. The sample size was based on the recent Cochrane review of 25.3% of median prevalence rate for return of menses by 6 months postpartum in a recent Cochrane review by Van der Wijden and Manion.

REVIEWER COMMENT

Again the method section looks scanty and lacks details of the sampling method used. Though, it is stated that "all eligible and consenting women seen at the end of the first six weeks of their childbirth were recruited" (line 6 page 8), It is hard to believe that in a tertiary hospital that manages all issues relating to pregnant women from within and outside the state, only 371 postnatal mothers were recruited over a 3 and half year period. How was the recruitment done?

AUTHORs’ RESPONSE

Many thanks for the remarks. A number of reasons were responsible: 1. We used a strict criteria such that women who were less than six weeks (≤5th week) postpartum or women more than six weeks (≥7th week) postpartum at the time of interview were excluded, among other exclusion criteria. Secondly, there were multiple industrial actions that affected the patient during the study period that affected the frequency of recruitment of women. However, we remained focused till the end of the study period. We have also appended it as one of the limitations of the study as follows: Also, there was a long interval of data collection which may have introduced bias of double counting in the study.

REVIEWER COMMENT

Again, it is stated that participants "were recruited until the calculated sample size was reached". How then was it possible for the sample size of 347 to be overachieved (371)?

AUTHORs’ RESPONSE

Many thanks for the remarks. We totally agree with the perspective of the reviewer. We have now deleted the statement: were recruited until the calculated sample size was reached" Overall, 371 women were recruited.

REVIEWER COMMENT

The authors need to check whether they used multiple logistic regression or multivariate logistic regression. The results in Table 3 suggest a binary multiple logistic regression was used as opposed to multivariate logistic regression.

AUTHORs’ RESPONSE

Multiple logistic regression analysis applies when there is a single dichotomous outcome and more than one independent variable. The 'multiple' applies to the number of predictors that enter the model (or equivalently the design matrix) with a single outcome (Y response), while 'multivariate' refers to a matrix of response vectors. In multivariate logistic regression, we have multiple dependent variables and multiple independent variables. Therefore a multiple logistic regression was used. Thank you.

REVIEWER COMMENT

Results:

There are several inconsistencies in the results presented few are shown below:

Parity: Table 1 shows 253 women were multiparous, however, Table 2 shows 248 (100 + 148).

Booking: 283 booked (table 1) compared 296 (table 2)

Social Class: 200 high (table 1) compared to 76 (Table 2)

AUTHORs’ RESPONSE

We regret the inconsistencies. The values in table 1 is now consistent with Table 2 in parity, booking status, social class and other parameters.

REVIEWER COMMENT

Discussion

On page 14 line 16 and 17, the authors indicated that "what matters most is the frequency and intensity of sucking of the infant which may be suboptimal in the majority of women studied", however, no data was presented in the results section on "frequency and intensity of sucking of the infants". What then is the basis for that assertion?

AUTHORs’ RESPONSE

We have adjusted the sentence to read as follows: Although this study did not assess the frequency and intensity of sucking of the infant, it may have been suboptimal in the majority of women studied. We have also appended it as one of the limitations of the study, as follows: Such studies should also assess the frequency and intensity of sucking of the infant in the breastfeeding women.

REVIEWER COMMENT

Reviewer #2: In general I think the paper has a strong justification, taking into account the low prevalence of family planning in this country and probably the high prevalence of unmet need for postpartum family planning. The conclusions could be useful to understand among the clinicians and patients the importance to increase prevention strategies to avoid unintended pregnancies during the postpartum period, especially in younger patients.

AUTHORs’ RESPONSE

Many thanks for the commendable comments.

REVIEWER COMMENT

Regarding the content, I have some minor comments:

• Page 4, line 2: Grammar mistake: To add “The” Puerperal mothers and remove “the” before nursing mothers

AUTHORs’ RESPONSE

Many thanks for the remarks. We have reflected the needed changes.

REVIEWER COMMENT

• Page 4, line 20: The acronym FP is not explain in the previous or in the subsequent paragraphs, it is well known that it refers to Family Planning but it is better to clarify.

AUTHORs’ RESPONSE

Many thanks for the remarks. FP is family planning. This has been corrected.

REVIEWER COMMENT

 Page 5, line 15-16: The statement “there have been previous studies on postpartum practices among parturients, none has studied the timing of initiation of menses“applies only for Nigeria or worldwide? If only applies for Nigeria, would be better to clarify.

AUTHORs’ RESPONSE

Many thanks for the remarks. We have appended the statements as follows: Although, there have been previous studies on postpartum practices among parturients, none has studied the timing of initiation of menses in a Nigerian population [1, 18].

REVIEWER COMMENT

• Page 8, line 21: What do you refer with booking status? Could you please clarify

AUTHORs’ RESPONSE

Many thanks for the remarks. Any woman who has attended and registered at least one antenatal clinic prior to delivery or labor is said to be booked.

REVIEWER COMMENT

• The lack of variable nutritional status is a limitation of this study because it has a direct influence on the amenorrhea postpartum period.

AUTHORs’ RESPONSE

Many thanks for the remarks. We agree with the perspective of the reviewer. Of course, we have appended it as a limitation of the study as follows: Our lack of information on nutritional status is a major limitation as this is likely to influence the amenorrhea in the postpartum period.

REVIEWER COMMENT

• Another limitation, is the small probability that some of these women have continued with postpartum loquios, and did not presented a real menstrual period, maybe would be good to clarify which is this probably according other studies.

AUTHORs’ RESPONSE

Many thanks for the remarks. We have similarly append as a limitations of the study as follows: Another limitation, is the small probability that some of these women may have continued with postpartum (delayed) bloody lochia, and did not presented a real menstrual period which follows the ovulation two weeks before, especially when most women were breastfeeding.

REVIEWER COMMENT

• Within the discussion I recommend to extend the analyses about the association between the manual removal of placenta and return of menses.

Thank you very much,

 AUTHORs’ RESPONSE

 We are happy to extend the analyses about the association between the manual removal of placenta and return of menstruation. We have now appended under discussion as follows: It is intriguing to observe the association between the manual removal of placenta and decreased incidence of puerperal menstruation. As long as we know, our study appears to be the first to report such potential association showing that women that underwent manual removal of placenta have significantly decreased incidence of puerperal menstruation. This finding is not surprising as manual removal of the placenta can also be a risk factor for acute postpartum endometritis [26]. However, when there is established puerperal menstruation in women with recent history of successfully managed retained placenta, the history of delayed hemorrhage or postpartum endometritis must be ruled out [26]. 

REVIEWER COMMENT

6. PLOS authors have the option to publish the peer review history of their article (what does this mean?). If published, this will include your full peer review and any attached files.

REVIEWER COMMENT

Do you want your identity to be public for this peer review? For information about this choice, including consent withdrawal, please see our Privacy Policy.

Reviewer #1: No

Reviewer #2: No

AUTHORs’ RESPONSE

Many thanks for your detailed review.

 AUTHORs’ RESPONSE

Many thanks for the commendations.

Thank you.

---

## [Decision Letter · Decision Letter 1]

17 Jun 2020

PONE-D-19-23514R1

Predictors of puerperal menstruation

PLOS ONE

Dear Dr. ELEJE,

Thank you for submitting your manuscript to PLOS ONE. After careful consideration, we feel that it has merit but does not fully meet PLOS ONE’s publication criteria as it currently stands. Therefore, we invite you to submit a revised version of the manuscript that addresses the points raised during the review process.

Both reviewers agree that the revised version of the manuscript has satisfied previous doubts and that it meets PLOS ONE's publicatioon criteria. After my evaluation of the manuscript, the following changes are required for acceptance.

- Abstract:  “childbirth” instead of “child birth”

- Last sentence of introduction: “This study therefore IS AIMED AT DETERMINING..”

- M&M, line 10: close parentheses:  (time at resumption of menses and coitus postpartum) .

- M&M Line 19: “validation. Firstly” instead of “validation: Firstly”. “Face validity of the questionnaire”

- page 10, line 24; remove  “given a ratio of 1:3”

- page 14, line 21, and following: “breastfeeding” instead of “breast feeding”

- page 15, line 3-4. Check spaces and -suboptimal

- page 15,line 20:   “counterpart” instead of “counter-part”

- discussion: “The main strength of this study was that this study appears to be the first study that attempts to describe puerperal menstruation. However, we could not by this study validate the determinants of other variables such as nutritional status of the women which could increase the risk of early return of menses [4, 5].”. I suggest to remove these sentences  and to begin the paragraph with: “The main limitation of our study is... “

- conclusion: remove “this study appears to be the first study to date that attempts..” and begin the paragraph with “Our data show that..”

- in table 3, indicate the ref value as suggested by Reviewer #1 (see below)

We look forward to receiving your revised manuscript.

Kind regards,

Alessio Paffoni, PhD

Academic Editor

PLOS ONE

Reviewers' comments:

Reviewer's Responses to Questions

**Comments to the Author**

1. If the authors have adequately addressed your comments raised in a previous round of review and you feel that this manuscript is now acceptable for publication, you may indicate that here to bypass the “Comments to the Author” section, enter your conflict of interest statement in the “Confidential to Editor” section, and submit your "Accept" recommendation.

Reviewer #1: All comments have been addressed

Reviewer #2: All comments have been addressed

2. Is the manuscript technically sound, and do the data support the conclusions?

Reviewer #1: Yes

Reviewer #2: Yes

3. Has the statistical analysis been performed appropriately and rigorously? 

Reviewer #1: Yes

Reviewer #2: Yes

4. Have the authors made all data underlying the findings in their manuscript fully available?

Reviewer #1: Yes

Reviewer #2: Yes

5. Is the manuscript presented in an intelligible fashion and written in standard English?

Reviewer #1: Yes

Reviewer #2: Yes

6. Review Comments to the Author

Reviewer #1: In Table 3, the reference should be stated for each of the independent variables. This will make it easier to interpret the odds ratios.

Reviewer #2: The authors adressed all the comments I did in the previous revision. They enlarged the information that was required. Many thanks.

7. PLOS authors have the option to publish the peer review history of their article (what does this mean?). If published, this will include your full peer review and any attached files.

Reviewer #1: No

Reviewer #2: Yes: Lina María Garnica Rosas

---

## [Author Response · Author response to Decision Letter 1]

23 Jun 2020

23-06-2020

From 

Corresponding author (George Eleje)

To 

Editor

PLOS ONE 

Dear editors:

Re: Submission of Response to Editors and Reviewers’ comments on Manuscript ID (Submission ID: PONE-D-19-23514R1) entitled "Predictors of puerperal menstruation”.

Please find enclosed a point-by-point response to the comments by the editors. We hope that the editors and reviewers will find the revisions acceptable.

Dear Dr. ELEJE,

Thank you for submitting your manuscript to PLOS ONE. After careful consideration, we feel that it has merit but does not fully meet PLOS ONE’s publication criteria as it currently stands. Therefore, we invite you to submit a revised version of the manuscript that addresses the points raised during the review process.

Both reviewers agree that the revised version of the manuscript has satisfied previous doubts and that it meets PLOS ONE's publication criteria. After my evaluation of the manuscript, the following changes are required for acceptance.

- Abstract: “childbirth” instead of “child birth”

AUTHORS’ RESPONSE

As suggested, we have now effected the changes needed. 

- Last sentence of introduction: “This study therefore IS AIMED AT DETERMINING..”

AUTHORS’ RESPONSE

As suggested, we have now effected the changes needed. 

- M&M, line 10: close parentheses: (time at resumption of menses and coitus postpartum) .

AUTHORS’ RESPONSE

As suggested, we have now effected the changes needed. 

- M&M Line 19: “validation. Firstly” instead of “validation: Firstly”. “Face validity of the questionnaire”

AUTHORS’ RESPONSE

As suggested, we have now effected the changes needed. 

- page 10, line 24; remove “given a ratio of 1:3”

AUTHORS’ RESPONSE

As suggested, we have now effected the changes needed. 

- page 14, line 21, and following: “breastfeeding” instead of “breast feeding”

AUTHORS’ RESPONSE

As suggested, we have now effected the changes needed. 

- page 15, line 3-4. Check spaces and –suboptimal

AUTHORS’ RESPONSE

As suggested, we have now effected the changes needed. 

- page 15,line 20: “counterpart” instead of “counter-part”

AUTHORS’ RESPONSE

As suggested, we have now effected the changes needed. 

- discussion: “The main strength of this study was that this study appears to be the first study that attempts to describe puerperal menstruation. However, we could not by this study validate the determinants of other variables such as nutritional status of the women which could increase the risk of early return of menses [4, 5].”. I suggest to remove these sentences and to begin the paragraph with: “The main limitation of our study is... “

- conclusion: remove “this study appears to be the first study to date that attempts..” and begin the paragraph with “Our data show that..”

AUTHORS’ RESPONSE

As suggested, we have now effected the changes needed. For instance: We have concluded as follows: In conclusion, our data show that approximately one third of the parturients achieve puerperal menstruation and the significant associated risk factors include younger age, multiparity, early use of family planning, high socio-economic status, ‘exclusive breastfeeding’ and manual removal of placenta. Of these, the major predictor was early initiation of family planning, while exclusive breastfeeding with manual removal of placenta a major protective factor. These study findings could be useful to understand among the clinicians and patients the importance to increase prevention strategies aimed at avoiding unintended pregnancies during the postpartum period, especially in younger women. This study has raised in¬teresting issues and requires further investigation to better understand the mechanism of puerperal menstruation.

- in table 3, indicate the ref value as suggested by Reviewer #1 (see below)

AUTHORS’ RESPONSE

As suggested, we have now indicated the reference value (Reference) as needed in table 3. 

We look forward to receiving your revised manuscript.

Kind regards,

Alessio Paffoni, PhD

Academic Editor

PLOS ONE

Reviewers' comments:

Reviewer's Responses to Questions

Comments to the Author

1. If the authors have adequately addressed your comments raised in a previous round of review and you feel that this manuscript is now acceptable for publication, you may indicate that here to bypass the “Comments to the Author” section, enter your conflict of interest statement in the “Confidential to Editor” section, and submit your "Accept" recommendation.

Reviewer #1: All comments have been addressed

Reviewer #2: All comments have been addressed

AUTHORS’ RESPONSE

Thank you.

2. Is the manuscript technically sound, and do the data support the conclusions?

Reviewer #1: Yes

Reviewer #2: Yes

AUTHORS’ RESPONSE

Thank you.

3. Has the statistical analysis been performed appropriately and rigorously?

Reviewer #1: Yes

Reviewer #2: Yes 

AUTHORS’ RESPONSE

Thank you.

4. Have the authors made all data underlying the findings in their manuscript fully available?

Reviewer #1: Yes

Reviewer #2: Yes 

AUTHORS’ RESPONSE

Thank you.

5. Is the manuscript presented in an intelligible fashion and written in standard English?

Reviewer #1: Yes

Reviewer #2: Yes

 AUTHORS’ RESPONSE

Thank you.

6. Review Comments to the Author

Reviewer #1: In Table 3, the reference should be stated for each of the independent variables. This will make it easier to interpret the odds ratios.

Reviewer #2: The authors adressed all the comments I did in the previous revision. They enlarged the information that was required. Many thanks.

AUTHORS’ RESPONSE

 As suggested, we have now indicated the reference value (Reference) as needed in table 3. 

7. PLOS authors have the option to publish the peer review history of their article (what does this mean?). If published, this will include your full peer review and any attached files.

Do you want your identity to be public for this peer review? For information about this choice, including consent withdrawal, please see our Privacy Policy.

Reviewer #1: No

Reviewer #2: Yes: Lina María Garnica Rosas

AUTHORS’ RESPONSE

Thank you.

---

## [Editor Report · Decision Letter 2]

25 Jun 2020

Predictors of puerperal menstruation

PONE-D-19-23514R2

Dear Dr. ELEJE,

We’re pleased to inform you that your manuscript has been judged scientifically suitable for publication and will be formally accepted for publication once it meets all outstanding technical requirements.

Kind regards,

Alessio Paffoni, PhD

Academic Editor

PLOS ONE

---

## [Editor Report · Acceptance letter]

30 Jun 2020

PONE-D-19-23514R2 

Predictors of puerperal menstruation 

Dear Dr. ELEJE:

I'm pleased to inform you that your manuscript has been deemed suitable for publication in PLOS ONE. Congratulations! Your manuscript is now with our production department. 

Kind regards, 

on behalf of

Dr. Alessio Paffoni 

Academic Editor

PLOS ONE